

# Oil and natural gas rents and CO$_2$ emissions nexus in MENA: spatial analysis

Haider Mahmood[1], Najia Saqib[2], Anass Hamadelneel Adow[3] and Muzaffar Abbas[4]

[1] Department of Finance, College of Business Administration, Prince Sattam bin Abdulaziz University, Alkharj, Saudi Arabia
[2] Department of Finance, College of Business Administration, Prince Sultan University, Riyadh, Saudi Arabia
[3] Department of Accounting, College of Business Administration, Prince Sattam Bin Abdulaziz University, Alkharj, Saudi Arabia
[4] Department of Business Administration, Community College, Prince Sattam Bin Abdulaziz University, Alkharj, Saudi Arabia

## ABSTRACT

**Background:** Oil rents (OR) and natural gas rents (NGR) have significant contributions to the income of the Middle East and North Africa (MENA) economies and may increase emissions. Moreover, spatial autocorrelation is expected in carbon dioxide (CO$_2$) emissions due to the geographically closed economies in the MENA region. Thus, we examine the impact of OR and NGR on CO$_2$ emissions caring spatial dimensions and analyze the environmental Kuznets curve (EKC).
**Methods:** We apply the spatial Durbin model technique on the effects of OR, NGR, and economic growth on CO$_2$ emissions in 17 MENA nations from 2000–2019, *i.e.*, Algeria, Bahrain, Egypt, Iran, Iraq, Israel, Jordan, Kuwait, Libya, Morocco, Oman, Qatar, Saudi Arabia, Syria, Tunisia, the United Arab Emirates (UAE), and Yemen. Moreover, diagnostic tests are applied to reach the most appropriate spatial specification and to have the most robust results.
**Results:** The results disclose that CO$_2$ emissions have spillovers and emissions of any country can damage the environment of neighboring countries. The EKC is corroborated with a turning point of 38,698 constant 2015 US dollars. Israel and Qatar are in 2$^{nd}$ phase of the EKC, and 15 MENA economies are in 1$^{st}$ stage. Thus, the economic expansion of most economies has ecological concerns. The effect of natural gas rents is found statistically insignificant. Oil rents have minute negative effects on emissions of local economies with an elasticity coefficient of −0.2117. Nevertheless, these have a positive indirect effect with an elasticity coefficient of 0.5328. Thus, the net effect of oil rents is positive. One percent increase in oil rents could accelerate 0.3211% of emissions. Thus, we suggest the MENA countries reduce reliance on oil rents in their income to avoid the negative environmental effects of the oil sector.

Corresponding author
Haider Mahmood,
haidermahmood@hotmail.com

## INTRODUCTION

Most Middle East and North Africa (MENA) nations have huge oil and natural gas reserves in their economies and their income is heavily dependent on oil rents. For instance, the average oil rents during the period 2000–2019 are 21.22%, 16.16%, 23.19%, 48.24%, 46.42%, 47.24%, 34.87%, 25.86%, 38.39%, 20.24%, and 22.14% of Gross Domestic Product (GDP) in Algeria, Bahrain, Iran, Iraq, Kuwait, Libya, Oman, Qatar, Kingdom of Saudi Arabia (KSA), the UAE, and Yemen, respectively. Moreover, the average natural rents during the period 2000–2019 are lesser than 6% in all MENA economies (*World Bank, 2022*). In the year 2020, six MENA economies are standing among the top 10 polluters in the globe as per Carbon Dioxide ($CO_2$) emissions *per capita* and are responsible for 8% of global $CO_2$ emissions (*Global Carbon Atlas, 2022*). As per the Paris Agreement, some MENA countries are expanding renewable energy consumption (REC) to avoid non-REC, but the transformation is slow except for Morocco and Jordan. Moreover, most MENA countries have a miserable share of REC in the entire energy mix (*Timmerberg et al., 2019*).

The above discussion moves the attention to testing the impact of oil rents (OR) and natural gas rents (NGR) on emissions in the high oil and natural gas-dependent MENA economies. However, the MENA literature could not test this relationship so far. Nevertheless, *Al-Mulali (2011)* analyzed and found the casual relationships between oil usage and $CO_2$ emissions. Moreover, some literature differentiated the impact of REC and non-REC on emissions. For example, some studies investigated the MENA panel and observed the positive impact of non-REC on emissions (*Farhani & Shahbaz, 2014*; *Alharthi, Dogan & Taskin, 2021*; *Omri & Saidi, 2022*). Further, the literature corroborated the negative impact of REC on emissions (*Omri & Saidi, 2022*; *Kahia, Ben Jebli & Belloumi, 2019*; *Alharthi, Dogan & Taskin, 2021*; *Charfeddine & Kahia, 2019*). *Farhani & Shahbaz (2014)* also substantiated the positive impact of REC on $CO_2$ emissions. Thus, the role of REC in reducing emissions is inconclusive. Moreover, some studies investigated and found the positive effect of aggregate Energy Consumption (EC) on emissions (*Al-Mulali et al., 2013*; *Omri, 2013*; *Abdallh & Abugamos, 2017*; *Ekwueme & Zoaka, 2020*).

The above-reported studies worked on EC variables. However, studies did not explore the impact of natural resources rents (NRR) on emissions. Resource rents are defined as the difference between revenues and the cost of production of resources (*World Bank, 2022*). So, natural resource rents represent the natural resource sector's value added to the GDP of the country. Thus, NRR is supporting the income of resource-producing economies. Moreover, oil and natural gas rents show the economic dependence of such economies on these natural resources, and the production of such resources could have environmental consequences. *Abbass, Kumar & El-Gendy (2018)* investigate and find that the energy sector is responsible for most emissions in the MENA region. Thus, oil and natural gas production and their rents in this region could cause environmental damage. The present study is different altogether from the MENA region literature. The past MENA literature could only focus on the role of the consumption of energy on emissions.

Moreover, the testing of NGR is scant in global literature. Thus, we contribute to both MENA and global literature.

We also include the spatial dimension in the relationship among OR, NGR, and $CO_2$ emissions. $CO_2$ emissions are global emissions (*Mahmood et al., 2023*) and emissions from one country could also pollute the neighboring and surrounding countries. Thus, spillovers of $CO_2$ emissions cannot be ignored. Moreover, *Bockstael (1996)* strongly recommended doing spatial analyses in the ecological studies of a region due to the common landscape and environmental and trading policies of a region. Thus, ignoring spatial dimensions in environmental studies in a region can produce biased results and misleading conclusions (*Maddison, 2007*), if a model statistically contains spatial autocorrelation (*Anselin, Le Gallo & Jayet, 2008*). Keeping in mind these arguments, the present study investigates the spatial effects of OR and NGR on $CO_2$ emissions in 17 MENA economies from 2000-2019. The maximum number of countries and time series from the MENA region are utilized as per the availability of data. The countries' sample include Algeria, Bahrain, Egypt, Iran, Iraq, Israel, Jordan, Kuwait, Libya, Morocco, Oman, Qatar, Saudi Arabia, Syria, Tunisia, the United Arab Emirates (UAE), and Yemen.

## LITERATURE REVIEW

The objective of this article is to investigate the effect of economic growth and oil and natural gas rents on $CO_2$ emissions in MENA countries. There is no single study in the investigation of the relationship between OR, NGR, and $CO_2$ emissions in the MENA region. Further, there is limited literature on testing the effects of NRR on pollution emissions in other regions or countries. So, we discuss literature in three dimensions. The first dimension will cover the effect of NRR on emissions in different regions or countries. The second dimension will discuss the environmental effects of the consumption of different energy sources. The third dimension will cover the environmental literature on the MENA region.

We start with the literature on testing the effect of NRR on emissions. For instance, *Ozturk (2017)* investigated nine Latin American economies from 1975–2013 by using seemingly unrelated regression and found that oil rents could not affect $CO_2$ emissions. Moreover, oil rents had a positive relationship with economic progress in Ecuador and Venezuela and a negative relationship in Argentina. Moreover, nuclear energy increased economic progress in Colombia, Peru, and Venezuela. *Bekun, Alola & Sarkodie (2019)* analyzed European Union (EU) countries by using Pooled Mean Group (PMG) from 1996–2014 and found that NRR, non-REC, and economic expansion boosted $CO_2$ emissions and REC reduced them. Moreover, feedback effects were found between NRR and economic expansion and between REC, non-REC, and economic expansion.

*Wang et al. (2020)* explored G7 nations by using Autoregressive Distributive Lag (ARDL) technique from 1996–2017 and corroborated that globalization, financial market development (FMD), and NRR increased $CO_2$ emissions. Nevertheless, agricultural activities lowered emissions. *Danish (2020)* explored the global economy from 1990–2013 by using ARDL and found that water productivity, NRR, and foreign trade increased $CO_2$ emissions. Moreover, the feedback effect was found between water productivity and

emissions. *Adedoyin et al. (2020)* examined Brazil, Russia, India, China, and South Africa (BRICS) from 1990–2014 by using PMG and found surprising results. Coal consumption and rents reduced $CO_2$ emissions and coal regulation increased $CO_2$ emissions.

*Joshua & Bekun (2020)* used ARDL and causality tests and found that coal consumption and economic expansion had feedback effects on each other in South Africa from 1970–2017. Moreover, NRR increased emissions. *Agboola, Bekun & Joshua (2021)* applied causality tests and discovered that EC, NRR, OR, and capital formation increased $CO_2$ emissions in KSA from 1970–2016. *Nwani & Adams (2021)* examined 93 economies and found that NRR boosted $CO_2$ emissions during 1995–2017 in countries having weak governance and reduced emissions in upper-high-governance economies. *Tufail et al. (2021)* probed the Organization for Economic Co-operation and Development (OECD) from 1990–2018 by using Cross-sectional Dependence (CD)-ARDL and found that fiscal decentralization and institutional quality helped to reduce $CO_2$ emissions. However, NRR and GDP increased emissions.

Considering spatial dimensions, *Mahmood & Furqan (2021)* probed Gulf Cooperation Council (GCC) economies from 1980–2014 by using Spatial Durbin Model (SDM) and found the EKC in $CO_2$, Methane ($CH_4$), Nitrous Oxide ($N_2O$), and Greenhouse Gas (GHG) emissions models. OR increased $CO_2$ and $N_2O$. FMD reduced all emissions and Foreign Direct Investment (FDI) reduced $CO_2$ and $CH_4$. EC increased $CO_2$ and $N_2O$. Moreover, all investigated variables showed spatial spillovers on some investigated emissions. *Shen et al. (2021)* studied 30 Chinese provinces by using CD-ARDL and substantiated that EC, NRR, and FMD increased $CO_2$ emissions during 1995–2017. However, green investment reduced emissions.

*Mahmood & Saqib (2022)* analyzed 13 organization of the petroleum exporting countries from 1970–2019 in asymmetrical settings of nonlinear ARDL. The EKC was substantiated in five countries in the long term and three economies in the short term. Increasing OR increased $CO_2$ emissions in eight countries and decreased emissions in three countries in the long run. Decreasing OR reduced $CO_2$ emissions in four countries and expanded emissions in one country. In the short-term results, increasing OR increased $CO_2$ emissions in four countries and decreased emissions in two countries. Moreover, decreasing OR lowered $CO_2$ emissions in four countries. *Royal, Singh & Chander (2022)* investigated G7 countries from 1971–2019 employing Fully Modified Ordinary Least Square (FMOLS) and found that oil prices and rents increased REC in the panel. However, FDI could not encourage REC. *Saqib, Duran & Hashmi (2022)* explored the GCC countries from 1993–2019 by using CD techniques and stated that economic expansion, NRR, and non-REC increased emissions. Financial deepening and REC reduced $CO_2$ emissions. The EKC was not validated.

*Huang & Guo (2022)* explored 30 Chinese provinces from 1995–2017 by using ARDL and found that NRR, transportation, innovation, FMD, and energy investment increased carbon emissions. *Gyamfi (2022)* investigated Sub-Saharan Africa from 1990–2018 by using CD techniques and found that FDI, economic expansion, NRR, and urbanization increased consumption-based emissions. *Adebayo et al. (2023)* showed in their article that many factors including REC, NRR, and technological innovation had a critical role to play

in BRICS $CO_2$ emissions trends. The authors used data from 1990–2019 and applied the ARDL model. The results showed that over a long period, technological innovations helped reduce emission levels. Additionally, REC and NRR reduced these emission levels. Technological innovation interacting with the other two variables of REC and NRR also reduced pollution in the region. Their analysis showed crucial policy implications in order to control emission levels by investing in more technological development and related factors.

*Luo et al. (2023)* investigated Asian economies from 1990–2018 and found that NNR in low-income countries increased emissions. Moreover, economic growth increased emissions in low- and middle-income economies and reduced in high-income economies. *Peng et al. (2023)* examined the effect of rents from different resources on $CO_2$ emissions and found that coal, oil, forest, and mineral rents increased $CO_2$ emissions. However, NGR, urbanization, and REC reduced $CO_2$ emissions. *Fu et al. (2023)* explored BRICS from 1990–2019 and found that NNR and expansionary fiscal policy increased emissions. However, REC and monetary contractions reduced emissions. *Nwani et al. (2023)* examined 45 developing economies from 1995–2017 and substantiated the EKC. Further, NNR and energy intensity raised $CO_2$ emissions. *Huang, Sadiq & Chien (2023)* analyzed the United States from 1995–2015 and found that NNR, FMD, and urbanization increased carbon emissions.

After discussions on the environmental effects of NRR, we exhibit literature on the effects of various energy sources on emissions. *Lim, Lim & Yoo (2014)* analyzed the Philippines from 1965–2012 by using causality analysis and found feedback between oil usage and $CO_2$ emission and between oil usage and economic expansion. *Alkhathlan & Javid (2015)* analyzed KSA and substantiated that aggregate oil and transport oil usage accelerated $CO_2$ emissions during 1971–2013. Moreover, economic expansion and its square increased $CO_2$ emissions in a nonlinear analysis. *Bildirici & Bakirtas (2016)* examined BRICS and Turkey from 1969–2011 by using ARDL and found that coal and oil usage caused $CO_2$ emissions. Moreover, the feedback effect between both energy sources and emissions was reported.

*Mensah et al. (2019)* analyzed twenty-two African countries from 1990–2015 by using CD techniques and found the feedback effects among fossil fuel use, emissions, and economic expansion. Moreover, oil prices also caused fossil fuel use, emissions, and income. *Saboori, Rasoulinezhad & Sung (2017)* investigated three economies from 1980–2013 employing a cointegration test and found that oil usage caused South Korean emissions and caused economic expansion in China and Japan. *Szetela et al. (2022)* examined the 10 resource-rich economies from 2000–2015 and found that REC reduced $CO_2$ emissions with a condition of good governance in the countries. Moreover, the EKC was also validated.

*Mahmood (2022)* examined GCC economies by using nonlinear ARDL and stated that oil and natural gas usage boosted $CO_2$ emissions during 1975–2019. The environmental influence of oil usage was found to be more than natural gas. *Belucio et al. (2022)* studied Europe from 1993–2018 by using ARDL. The authors stated that both oil and natural gas usage raised $CO_2$ emissions. The coefficient of oil consumption on emission was lesser

than 1, which indicated the energy efficiency gains. Moreover, the effect of oil consumption was 6.7 times greater than natural gas and REC reduced $CO_2$ emissions. *Aslam et al. (2022)* investigated Malaysia from 1971–2016 by using ARDL and found that liquid fuel consumption, trade openness, industrial sector, and economic performance increased $CO_2$ emissions. Moreover, a feedback effect was reported between liquid fuel usage and trade.

There is no research investigating the impact of NRR on emissions in MENA panel. However, we discuss the studies considering the impact of EC and other macroeconomic factors on emissions. *Al-Mulali (2011)* explored the MENA economies from 1980–2009 and found feedback among oil usage, $CO_2$ emission, and economic expansion. *Omri (2013)* examined 14 MENA countries from 1990–2011 by using a causality test and found the feedback effect between economic expansion and emissions. Moreover, a two-way causality was reported in EC and economic expansion. However, the one-way effect is reported from EC to $CO_2$ emissions. *Al-Mulali et al. (2013)* probed the MENA region from 1980–2009 by using a causality test and found that urbanization and energy consumption shared feedback effects with $CO_2$ emissions in the region.

*Farhani & Shahbaz (2014)* investigated 10 MENA economies from 1980–2009 by using FMOLS and substantiated the EKC. Moreover, both REC and non-REC accelerated $CO_2$ emissions. *Abdallh & Abugamos (2017)* probed 20 MENA economies from 1980–2014 by using semi-parametric techniques and found that urbanization had a minute impact on emissions in nonlinear analyses. Moreover, economic expansion and energy consumption were responsible for higher emissions. *Charfeddine & Kahia (2019)* examined and found that REC and FMD had a minute effect on emissions in 24 MENA economies from 1980–2015, which showed a weak position of these economies to have pleasant environmental effects from REC and FMD. *Kahia, Ben Jebli & Belloumi (2019)* studied 12 MENA economies from 1980–2012 by using cointegration and causality analyses and found that economic expansion increased $CO_2$ emissions. REC, foreign trade, and FDI helped reduce $CO_2$ emissions. *Ekwueme & Zoaka (2020)* investigated 10 MENA countries from 1970–2017 and found that FMD decreased the effusion of $CO_2$ emissions. Nevertheless, trade openness and EC raised the effusion of $CO_2$ emissions.

*Alharthi, Dogan & Taskin (2021)* employed quantile regression and found that REC reduced $CO_2$ emissions and its effect increased over higher quantiles in the MENA region from 1990–2015. Non-REC increased $CO_2$ emissions and its effect was decreased over higher quantiles. The EKC was also confirmed in the region. *Ben Cheikh & Ben Zaied (2021)* examined the MENA region by using threshold regression and found that income reduced emissions, which was due to changing energy mix towards renewable and low-carbon technologies. *Omri & Saidi (2022)* investigated 14 MENA economies by using FMOLS and found that REC reduced $CO_2$ emissions and non-REC and industrial sectors accelerated $CO_2$ emissions. Moreover, a feedback effect was stated among non-REC, REC, and emissions and between the agriculture sector and $CO_2$ emissions.

Some literature could not include energy variables but tested the effect of other macroeconomic variables on emissions. *Guoyan et al. (2022)* analyzed the MENA region by using panel smooth transition regression (PSTR) and found that FDI helped in minimizing $CO_2$ emissions during 1995–2016. *Ben Lahouel et al. (2022)* explored 16

MENA economies from 1990–2019 by using PSTR and found that information technology could help reduce $CO_2$ emissions after a threshold point. *Gorus & Aydin (2019)* examined eight MENA economies from 1975–2014 and could not find any indication of causality between economic expansion and $CO_2$ emissions. Thus, economic growth could not be achieved at a level to support $CO_2$ emissions reduction. *Sinha et al. (2020)* probed MENA economies from 1990–2017 by quantile-on-quantile regression and found feedback effects between technological progress and ambient air pollution.

The reviewed literature finds a positive impact of NRR on emissions in most studies (*Bekun, Alola & Sarkodie, 2019*; *Wang et al., 2020*; *Danish, 2020*; *Joshua & Bekun, 2020*; *Tufail et al., 2021*; *Mahmood & Furqan, 2021*; *Shen et al., 2021*; *Saqib, Duran & Hashmi, 2022*; *Huang & Guo, 2022*; *Luo et al., 2023*; *Fu et al., 2023*; *Nwani et al., 2023*; *Huang, Sadiq & Chien, 2023*). However, the negative effect of NRR is also corroborated by a few studies (*Adedoyin et al., 2020*; *Nwani & Adams, 2021*; *Mahmood & Saqib, 2022*; *Adebayo et al., 2023*). Thus, the testing of the effect of NRR on emissions is an empirical question for any country or region. However, the effects of OR and NGR on $CO_2$ emissions are not tested in MENA economies. This present research is an effort to examine this question in the MENA region to claim a contribution to the MENA literature. Moreover, aggregate NRR is mostly explored in the literature, and testing the effect of NGR is scant. Thus, the testing of the separate effects of OR and NGR on emissions also ensures a contribution to the globe and the MENA literature.

## METHODS

The goal of this research is to assess the role of economic growth, OR, and NGR on $CO_2$ emissions in resource-abundant MENA economies. Here, we cannot ignore the testing of the EKC, which explains the nonlinear effect of economic expansion on emissions (*Grossman & Krueger, 1991*). For instance, energy is used extensively in the first stage of growth. Thus, economic growth may increase emissions by a scale effect. Later, economic growth would increase the demand for a clean environment and emissions may be reduced because of technique and/or composition effects in an economy (*Grossman & Krueger, 1991*; *Vukina, Beghin & Solakoglu, 1999*). Therefore, economic growth may increase emissions in the first phase of the EKC and could reduce emissions at a later stage. In this way, an inverted U-shaped effect of economic growth is expected on emissions, which is called the EKC hypothesis. Moreover, natural resources would also help in shaping the EKC. *Meadows et al. (1972)* claimed that the excessive use of natural resources could limit global growth as natural resources are limited on the globe. *Auty (1985)* did a pioneer study in this context, which corroborated an inverted U-shaped connection between economic expansion and metal use. Moreover, natural resources would accelerate emissions because energy and natural resources consumption would be used at the same rate in an early phase of growth (*Dinda, 2004*). Thus, OR and NGR would release emissions in the oil and natural gas-rich MENA countries. In addition, OR and NGR carry a significant proportion of the GDP of most MENA countries. Thus, increasing overall income from OR and NGR would increase economic activities and energy consumption, which is called the scale effect. Moreover, increasing income out of OR and NGR could also have the dominant

technique and/or composition effects. Thus, both OR and NGR could help in shaping the EKC. So, the present research adds both variables of OR and NGR in the model to test the EKC and the basic model is as follows:

$$CO2_{it} = f\left(Y_{it}, Y_{it}^2, OR_{it}, NGR_{it}\right) \qquad (1)$$

$CO2_{it}$ is the natural log of *per capita* $CO_2$. $Y_{it}$ is a natural log of *per capita* GDP (constant 2015 US $). $Y_{it}^2$ is the square of $Y_{it}$. The quadratic effect of economic growth is included in a model to examine the EKC hypothesis. The $Y_{it}$ and $Y_{it}^2$ are used in the model to capture the nonlinear effect of economic growth on $CO_2$ emissions. If $Y_{it}$ and $Y_{it}^2$ have positive and negative effects on $CO_2$ emissions, then the EKC can be validated with an inverted U-shaped effect of GDP *per capita* on $CO_2$ emissions. $OR_{it}$ and $NGR_{it}$ are the percentages of oil rents and natural gas rents in GDP, respectively. These variables are not converted in logarithms as both are already in percentages. *i* represents Algeria, Bahrain, Egypt, Iran, Iraq, Israel, Jordan, Kuwait, Libya, Morocco, Oman, Qatar, Saudi Arabia, Syria, Tunisia, the UAE, and Yemen. *t* shows the annual time sample of the period 2000–2019. Annual data from 17 MENA countries during 2000–2019 are obtained from *World Bank (2022)* on variables mentioned in Eq. (1). Country and time sample is selected based on data availability.

Equation (1) will be estimated by using the Fixed Effects (FE) model and the Likelihood-Ratio (LR) test will be applied to check the appropriateness of FE over pooled regression to check the most suitable model specification. Thereon, Lagrange Multiplier (LM) and LM robust tests will be utilized to test the possible spatial autocorrelation (*Debarsy & Ertur, 2010*). In pollution-related studies, spatial dimensions are expected (*Bockstael, 1996*). It is particularly important for the MENA region as the MENA countries are sharing almost the same landscape. Moreover, some MENA countries are also neighboring economies. Secondly, $CO_2$ emissions are global emissions, and $CO_2$ emissions in one country could affect the ecology of neighboring or nearby countries. If spatial autocorrelation is found significant in statistical analyses, then SDM is suggested to capture the spillovers (*Elhorst, 2012*). The Spatial Autoregressive (SAR) and Spatial Error Model (SEM) are nested in the SDM. Therefore, it is better to apply the SDM first. Later SDM can be compared with SAR and SEM to choose the best spatial specification. The SDM of Eq. (1) can be presented as follows:

$$\begin{aligned} CO2_{it} = {}& \alpha_{10} + \alpha_{11}Y_{it} + \alpha_{12}Y_{it}^2 + \alpha_{13}OR_{it} + \alpha_{14}NGR_{it} + \beta_{11}W.Y_{it} + \beta_{12}W.OR_{it} \\ & + \beta_{13}W.NGR_{it} + \delta_1 W.CO2_{it} + v_{1i} + u_{1t} + \vartheta_{1it} \end{aligned} \qquad (2)$$

$W$ is a 17*17-dimension matrix to capture the effect of distance between MENA countries. Thus, $W$ captures spillovers of all variables on $CO_2$ emissions in the nearby countries of the MENA region. *Kelejian & Prucha (2010)* suggested a method to normalize $W$ to have robust results and to capture spillovers of $CO_2$ emissions in neighboring countries. The time and countries' effects are captured in $v_{1i}$ and $u_{1t}$, respectively. The $\vartheta_{1it}$ is a normally distributed residual term of the model. Thereafter, *Elhorst (2010)* recommended the Wald test to verify the best spatial specification. A null hypothesis ($H_{01}$)

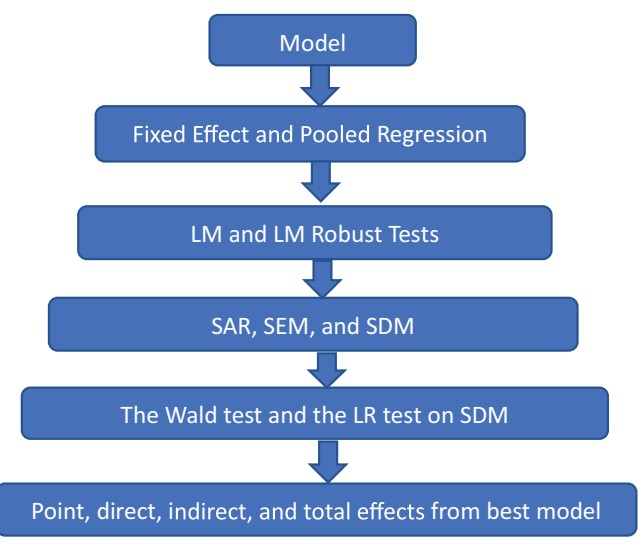

**Figure 1 Flow of methodology.**

of the Wald is $\beta_j = 0$, which will be tested to verify whether SDM tends to be converted to SAR specification or not. The SAR model is as follows:

$$CO2_{it} = \alpha_{20} + \alpha_{21}Y_{it} + \alpha_{22}Y_{it}^2 + \alpha_{23}OR_{it} + \alpha_{24}NGR_{it} + \delta_2 W.CO2_{it} + v_{2i} + u_{2t} + \vartheta_{2it} \quad (3)$$

The Wald test may also be applied on $H_{02}$: $\beta + \delta.\alpha = 0$, which may also be tested to confirm the validity of SDM over the SEM. The SEM model is as follows:

$$CO2_{it} = \alpha_0 + \alpha_{11}Y_{it} + \alpha_{12}Y_{it}^2 + \alpha_{13}OR_{it} + \alpha_{14}NGR_{it} + v_{3i} + u_{3t} + \vartheta_{3it} \quad (4)$$

$$\delta_3 W\vartheta_{3it} + \varepsilon_{it} = \theta_{it} \quad (5)$$

The rejection of both hypotheses may verify that the SDM is the best choice. Otherwise, the estimations may move toward SAR or SEM specifications. Figure 1 explains all procedures of methodology. For instance, the model will be regressed by pooled regressions and FE. Then, the LM and LM robust tests will be applied to test the existence of spatial autocorrelation in the model. If spatial autocorrelation is found statistically significant, then the SDM, SAR, and SEM models will be applied. Afterward, the LR test and the Wald test will be applied to SDM to test the most suitable spatial specification. Lastly, point, direct, indirect, and total effects will be estimated from the most suitable spatial specification.

## RESULTS AND DISCUSSION

We start with discussions of the spatial distribution of $CO_2$ emissions and oil and natural gas rents. Figure 2 shows the spatial distribution of an average oil rent percentage of GDP during 2000–2019 in 17 MENA countries. Israel and Jordon are neighboring countries having oil rents less than 1% of GDP. Egypt, Syria, and Tunisia are nearby countries having oil rents between 1–9.9% of GDP. Bahrain, Iran, Qatar, and the UAE are neighboring countries having oil rents between 10–29.9% of GDP. Iraq, Kuwait, Oman, and Saudi
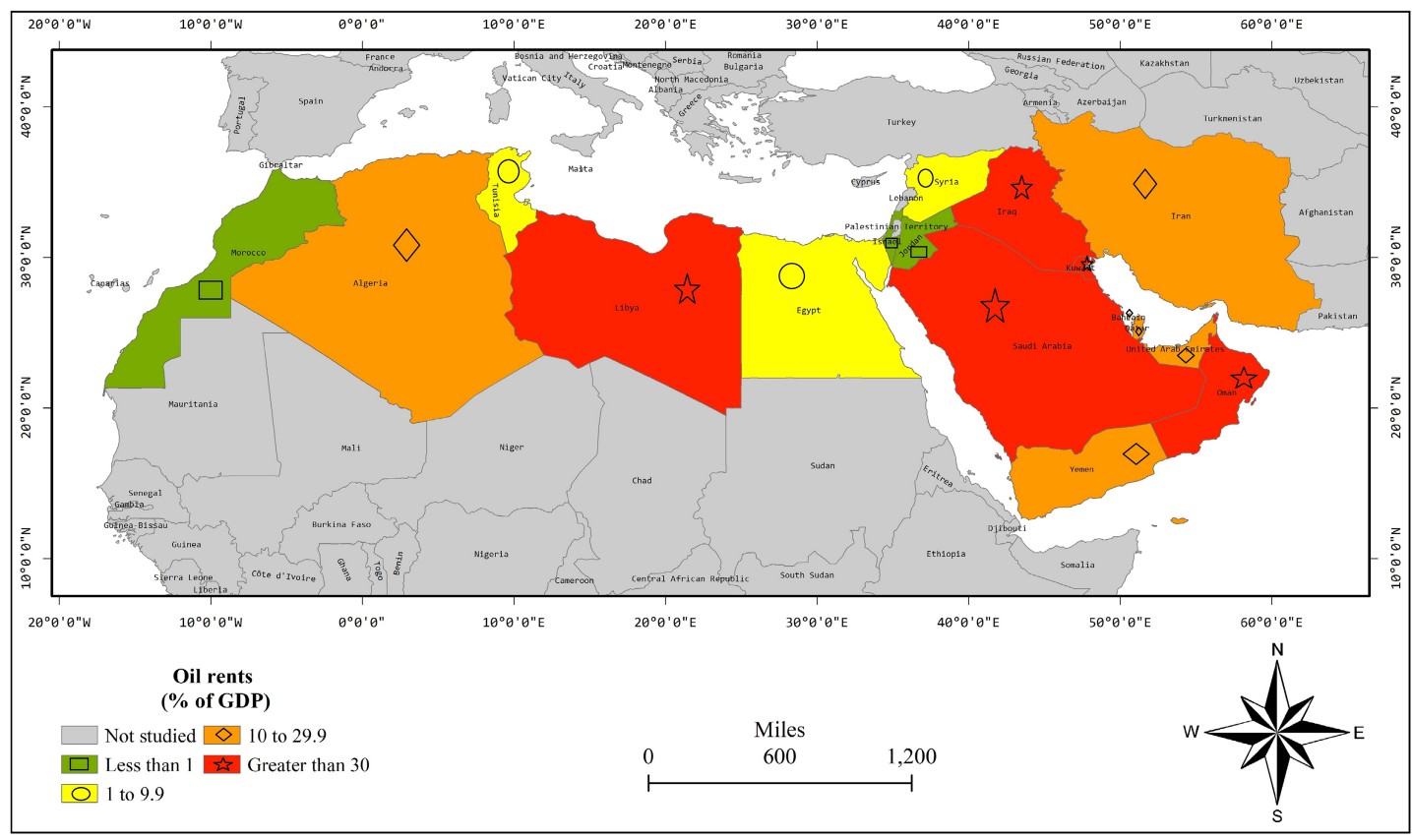

**Figure 2 OR percentage of GDP average of years 2000–2019.**

Arabia are neighboring countries having oil rents of more than 30% of GDP. Thus, Fig. 1 shows the spatial linkages in terms of oil rents in MENA countries.

Figure 3 shows the spatial distribution of an average natural gas rent percentage of GDP during 2000–2019. Iraq, Israel, Jordon, Kuwait, Saudi Arabia, and the UAE are neighboring countries having natural gas rents lesser than 1% of GDP. Moreover, Morocco and Tunisia are nearby countries with a similar range of natural gas rents. Egypt and Libya are neighboring countries having natural gas rents between 1–1.9% of GDP. Bahrain, Iran, and Oman are nearby countries having natural gas rents between 2–2.9% of GDP. Hence, Fig. 3 shows the spatial linkages in terms of NGR in MENA countries.

Figure 4 demonstrates the spatial distribution of an average of $CO_2$ emissions *per capita* during 2000–2019. Algeria, Egypt, Iran, Iraq, Israel, Jordan, Libya, Syria, and Tunisia are mostly neighboring countries and are located on a horizontal script in the map having $CO_2$ emissions between 2–9.9 metric tons *per capita*. Oman and Saudi Arabia are neighboring countries having $CO_2$ emissions between 10–19.9. Bahrain, Kuwait, Qatar, and the UAE are neighboring countries having $CO_2$ emissions of more than 20. Thus, Fig. 4 shows the spatial linkages in $CO_2$ emissions.

Table 1 exposes the results of FE without spatial dimensions. At first, we apply the LR test on FE and the results reject the $H_0$ for FE-country and FE-both specifications with a *p*-value lesser than 0.01. So, these both specifications are valid and superior over pooled
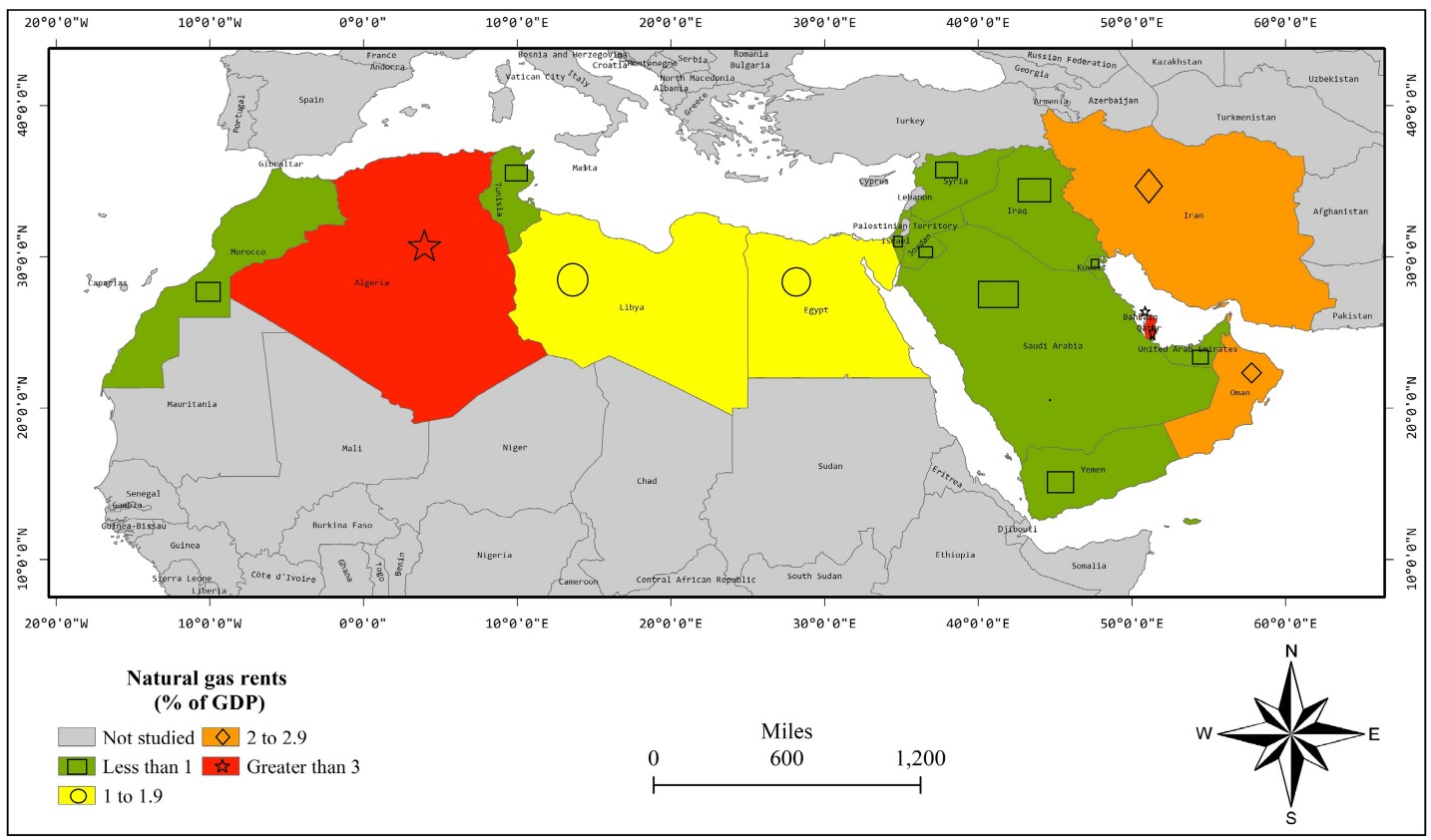

**Figure 3 NGR percentage of GDP average of years 2000–2019.**           

regression. However, the LR test could not reject $H_0$ for FE-time with a *p*-value = 0.992. Thus, it is not a valid specification. In both FE-country and FE-both, the EKC is substantiated by the positive and negative parameters of $Y_{it}$ and $Y_{it}^2$, respectively. Moreover, oil rents are decreasing $CO_2$ emissions in FE-both estimation and have insignificant effect in the FE-country model. Moreover, natural gas rents have insignificant effects in FE-country and FE-both models. Hereafter, we apply LM and LM robust tests as Figs. 2–4 show the spatial linkages, which reject the $H_0$ in all models with a *p*-value lesser than 0.01. Thus, FE models are biased as spatial autocorrelation is substantiated in all models (*Anselin, Le Gallo & Jayet, 2008*). Thus, we switch to the SDM estimation.

For completeness, we estimate SDM in FE and Random Effects (RE) with country and time effects. Then, we apply the Wald and LR tests on both models to test the suitability of the SDM. In Table 2, both tests reject $H_{01}$ and $H_{02}$ in both models with a *p*-value lesser than 0.01. So, SDM cannot reduce to SAR or SEM. Thus, SDM is the most suitable model specification. Afterward, the Hausman test is applied and $H_0$ is rejected with a *p*-value lesser than 0.01. Thus, SDM with FE specification may be preferred over RE and this model is chosen for interpretation. In SDM with FE specification, we ignore the interpretation of point estimates, which do not carry spatial effects. The coefficient of $W^*CO2_{it}$ is positive. Thus, increasing $CO_2$ emissions in one MENA country is increasing $CO_2$ emissions in the neighboring MENA countries by the spillover effects. In the same way, the coefficient of

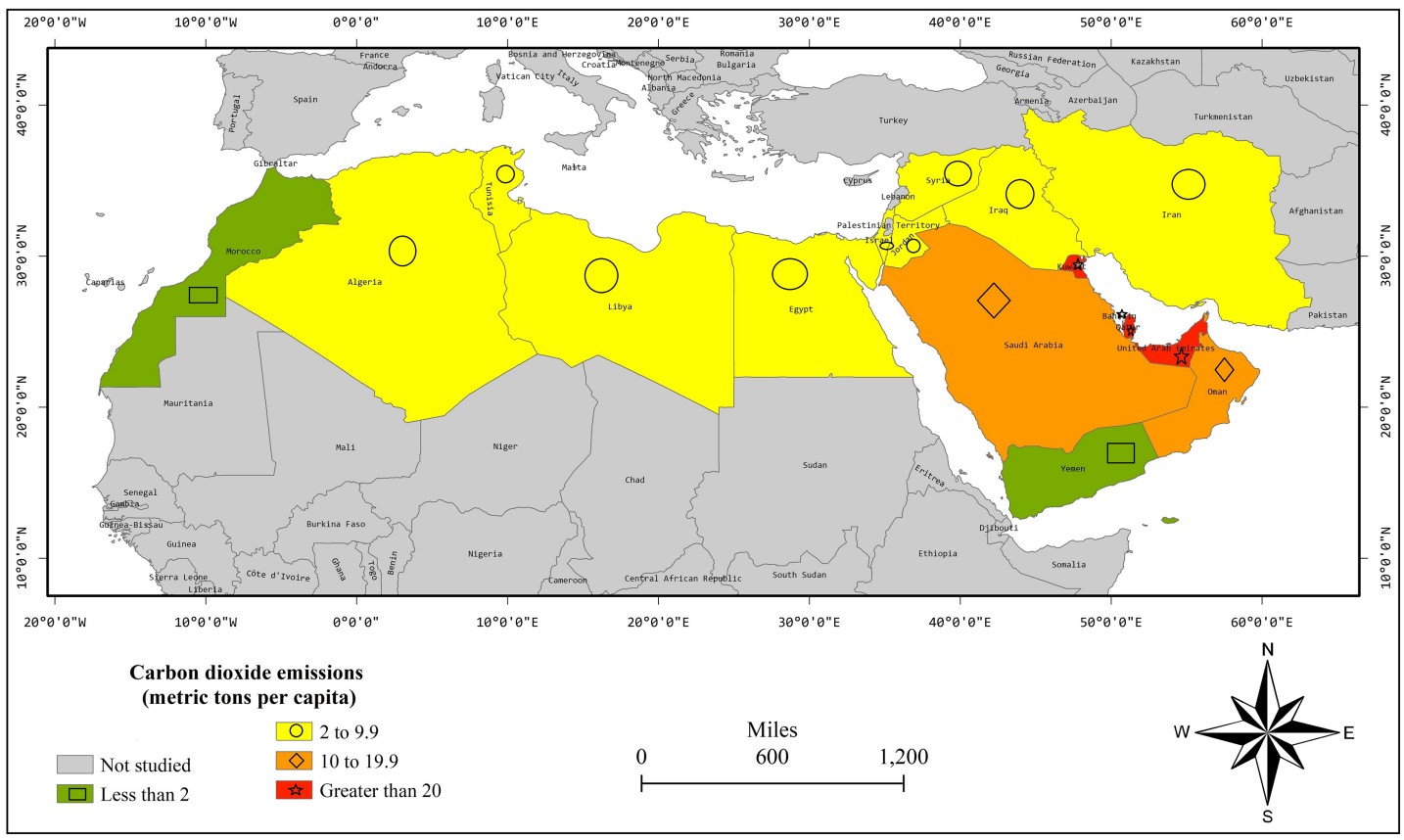

**Figure 4** CO$_2$ emission *per capita* average of years 2000–2019.

**Table 1 Non-spatial results.**

| Variables | Pooled regression | FE-country | FE-time | FE-both |
|---|---|---|---|---|
| $Y_{it}$ | 1.142 (0.001) | 3.516 (0.000) | 1.185 (0.001) | 3.726 (0.000) |
| $Y_{it}^2$ | −0.018 (0.326) | −0.157 (0.000) | −0.021 (0.276) | −0.166 (0.000) |
| $OR_{it}$ | 0.681 (0.000) | 0.068 (0.405) | 0.721 (0.000) | −0.226 (0.046) |
| $NGR_{it}$ | 9.222 (0.000) | −0.095 (0.911) | 10.203 (0.000) | 0.136 (0.899) |
| LM spatial lag | 776.224 0.000) | 441.045 (0.000) | 798.788 (0.000) | 419.290 (0.000) |
| Robust LM spatial lag | 15.366 (0.000) | 63.744 (0.000) | 10.972 (0.001) | 57.044 (0.000) |
| LM spatial error | 814.217(0.000) | 399.113 (0.000) | 837.734 (0.000) | 396.435 (0.000) |
| Robust LM spatial error | 53.360 (0.000) | 21.811 (0.000) | 49.918 (0.000) | 34.189 (0.000) |
| $\sigma^2$ | 0.134 | 0.012 | 0.145 | 0.011 |
| $R^2$ | 0.890 | 0.991 | 0.892 | 0.993 |
| LR test | | 864.680 (0.000) | 7.290 (0.992) | 900.810 (0.000) |

**Note:**
Probability values are presented in ().

$W^*OR_{it}$ is positive. Thus, increasing oil rents in one MENA country is increasing CO$_2$ emissions in neighboring countries. The rest spatial effects are statistically insignificant.

**Table 2 SDM results.**

| Variables | FE with both | RE with both |
|---|---|---|
| | Parameter (*p*-value) | Parameter (*p*-value) |
| Point estimates | | |
| $Y_{it}$ | 3.495 (0.000) | 3.506 (0.000) |
| $Y_{it}^2$ | −0.153 (0.000) | −0.156 (0.000) |
| $OR_{it}$ | −0.204 (0.041) | −0.227 (0.063) |
| $NGR_{it}$ | 1.043 (0.270) | −0.437 (0.693) |
| Direct estimates | | |
| $Y_{it}$ | 3.549 (0.000) | 3.578 (0.000) |
| $Y_{it}^2$ | −0.156 (0.000) | −0.159 (0.000) |
| $OR_{it}$ | −0.212 (0.032) | −0.249 (0.021) |
| $NGR_{it}$ | 1.002 (0.290) | −0.094 (0.929) |
| Indirect estimates | | |
| $Y_{it}$ | −1.172 (0.000) | −1.485 (0.000) |
| $Y_{it}^2$ | 0.043 (0.007) | 0.048 (0.015) |
| $OR_{it}$ | 0.533 (0.000) | 0.941 (0.228) |
| $NGR_{it}$ | 0.855 (0.559) | −9.136 (0.150) |
| Total estimates | | |
| $Y_{it}$ | 2.377 (0.000) | 2.094 (0.000) |
| $Y_{it}^2$ | −0.113 (0.000) | −0.111 (0.000) |
| $OR_{it}$ | 0.321 (0.001) | 0.691 (0.394) |
| $NGR_{it}$ | 1.857 (0.186) | −9.230 (0.165) |
| Weights | | |
| $W^*Y_{it}$ | −0.236 (0.171) | −0.523 (0.283) |
| $W^*OR_{it}$ | 0.646 (0.000) | 1.102 (0.310) |
| $W^*NGR_{it}$ | 1.627 (0.370) | −12.494 (0.131) |
| $W^*CO2_{it}$ | 0.394 (0.024) | 0.434 (0.039) |
| $R^2$ | 0.866 | 0.858 |
| $\sigma^2$ | 0.010 (0.000) | 0.009 (0.000) |
| Spatial lag-LR test | 21.180 (0.000) | 28.360 (0.000) |
| Spatial error-LR test | 35.380 (0.000) | 39.580 (0.000) |
| Spatial lag-Wald test | 21.890 (0.000) | 24.420 (0.00) |
| Spatial error-Wald test | 19.600 (0.000) | 11.660 (0.009) |
| Hausman test | 83.180 (0.000) | |

In the direct estimates, the parameters of $Y_{it}$ and $Y_{it}^2$ are positive and negative, respectively. Accordingly, an inverted U-shaped relationship is found between GDP *per capita* and $CO_2$ emissions. So, the EKC is corroborated in the local economies. Moreover, oil rents have a negative impact with a coefficient of −0.212. Thus, 1% increase in $OR_{it}$ is reducing emissions by 0.212%, which is a minute effect. Thus, oil rents are helping to reduce $CO_2$ emissions, but the local MENA economies are not mature enough in reducing $CO_2$ emissions at a large scale. The effect of $NGR_{it}$ is noticed as statistically insignificant.

Thus, NGR does not help reduce $CO_2$ emissions. However, these are not at least increasing $CO_2$ emissions. Thus, its environmental effect is neutral. This result corroborates the fact that oil rents are significantly higher than natural gas rents. For instance, the percentage of OR in GDP is more than 20% in 10 out of 17 MENA economies. On the other hand, the percentage of NGR in GDP is lesser than 6% in all MENA economies and lesser than 1% in most MENA economies.

In the indirect estimates, economic growth substantiates a U-shaped impact on $CO_2$ emissions with the negative and positive coefficients of $Y_{it}$ and $Y_{it}^2$, respectively. The EKC is not substantiated by the spillover effects. $OR_{it}$ carries a positive impact with a coefficient of 0.533 and 1% increasing $OR_{it}$ is increasing emissions by 0.533% in neighboring countries. Moreover, the spillover coefficient is greater than the direct effect coefficient. Consequently, the net impact of $OR_{it}$ is increasing $CO_2$ emissions in the MENA region. The spillover effect of $NGR_{it}$ is statistically insignificant. Thus, $NGR_{it}$ has no environmental spillover on the $CO_2$ emissions of neighboring countries. Moreover, the parameter of $W^*CO2_{it}$ is positive. Thus, increasing $CO_2$ in a MENA country is responsible for increasing emissions in neighboring economies as well.

In the total estimates, the coefficients of $Y_{it}$ and $Y_{it}^2$ are 2.377 and −0.113, respectively. The EKC is corroborated in the whole region, which is also substantiated by previous MENA literature (*Farhani & Shahbaz, 2014*; *Alharthi, Dogan & Taskin, 2021*). The turning point of this shape is approximately 38,698 constant 2015 US dollars ($e^{2.3768/2/0.1125}$) and Qatar and Israel are found in the 2nd phase. Israel is not much dependent on OR and NGR. Qatar has achieved the top *per capita* GDP among the MENA countries. Thus, economic expansions have pleasant environmental outcomes in Qatar and Israel. However, the rest of MENA economies are still in 1st phase of the EKC, and their economic expansions have environmental concerns. $OR_{it}$ has a positive effect with a coefficient of 0.321. So, 1% increasing $OR_{it}$ is raising emissions by 0.321% in the whole MENA region. Thus, increasing OR has environmental damage for the whole region. An average oil rent percentage of the GDP is more than 20% in 10 out of 17 investigated MENA countries. Out of which, five countries are carrying more than 30% of oil rents in GDP and three countries are carrying more than 40% of oil rents in GDP. Thus, the GDP of most MENA countries is highly dependent on oil rents. Thus, this dependence on the oil sector leads the MENA region toward environmental problems in terms of increasing $CO_2$ emissions as per our results. On the production side, oil extraction releases a huge amount of combustion gases including $CO_2$ emissions. Thus, oil production and rents have environmental concerns for the MENA region. This finding of the OR effect is also supported by *Agboola, Bekun & Joshua (2021)* in KSA, *Mahmood & Furqan (2021)* in GCC, and *Mahmood & Saqib (2022)* in OECD. The effect of NGR is statistically insignificant. Thus, NGR does not affect the environment, which is matched by the fact that the percentage of NGR in GDP is lesser than 6% in the case of all investigated MENA countries, in which most countries are carrying lesser than 1% of NGR in GDP. Thus, the dependence of the MENA region is not much significant on the natural gas sector. Therefore, NGR could not affect the environment of the MENA region significantly.

## CONCLUSION

The MENA countries are OR- and NGR-abundant economies, which would damage the environment. Moreover, these are geographically nearby countries, and their pollution emissions could have spillovers on each other. Thus, the present study investigates the EKC and the effects of OR and NGR on $CO_2$ emissions in the MENA region by using SDM. The results show that $CO_2$ emissions have spatial autocorrelation and these global emissions of one country are also damaging the environment of their nearby MENA countries. Thus, any economic growth and environmental policies should consider the spatial effect of $CO_2$ emissions to protect the MENA neighboring countries from emissions. Moreover, the EKC is substantiated with a turning point of 38,698 constant 2015 US dollars. Israel and Qatar are found in $2^{nd}$ stage of the EKC, and their economic expansions are helping in the reduction of $CO_2$ emissions. However, the remaining 15 MENA economies are in the $1^{st}$ stage and their economic growth has environmental concerns. OR has a minute negative impact on emissions in local economies with an elasticity coefficient of −0.212. Therefore, a 1% increase in OR may reduce 0.212% of $CO_2$ emissions in local economies. However, the spillovers of oil rents are positive and damaging the environment with an elasticity of 0.533. 1% increase in OR may increase 0.533% of $CO_2$ emissions in neighboring economies. Moreover, the net effect of oil rent is damaging the environment in the region, and 1% increasing OR is releasing 0.321% of $CO_2$ emissions. Therefore, MENA countries are advised to reduce the dependence of their economies on oil rents to save the environment from the oil sector. For this purpose, the government should focus on the growth of the non-oil sector to replace the oil sector. The effect of NGR is statistically insignificant. Thus, NGR has neither positive nor negative effects on the environment of the MENA region. Thus, natural gas production should be encouraged in MENA economies, which do not have at least any environmental problems as per our results.

MENA countries are oil and natural gas-abundant economies. The present study works on the MENA region and utilizes two types of natural resource rents. A future study may increase the scope of the research by increasing the sample countries. Moreover, types of natural resource rents may also be increased to show the effects of different natural resource rents on pollution emissions.

## ACKNOWLEDGEMENTS

We thank the anonymous reviewers and academic editor for their valuable comments.

### Funding

The authors received funding from the Deputyship for Research & Innovation, Ministry of Education in Saudi Arabia through project number (IF2/PSAU/2022/02/21824). The funders had no role in study design, data collection and analysis, decision to publish, or preparation of the manuscript.

## Grant Disclosures

The following grant information was disclosed by the authors:
Deputyship for Research & Innovation, Ministry of Education in Saudi Arabia: IF2/PSAU/2022/02/21824.

## Competing Interests

Haider Mahmood is an Academic Editor for PeerJ.

## Author Contributions

- Haider Mahmood conceived and designed the experiments, performed the experiments, analyzed the data, prepared figures and/or tables, authored or reviewed drafts of the article, and approved the final draft.
- Najia Saqib performed the experiments, prepared figures and/or tables, authored or reviewed drafts of the article, and approved the final draft.
- Anass Hamadelneel Adow performed the experiments, authored or reviewed drafts of the article, and approved the final draft.
- Muzaffar Abbas performed the experiments, authored or reviewed drafts of the article, and approved the final draft.

## Data Availability

   The raw data and codes are available in the Supplemental Files.

## Supplemental Information

Supplemental information for this article can be found online at http://dx.doi.org/10.7717/peerj.15708#supplemental-information.

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
