# Peer review of "Oil and natural gas rents and CO2 emissions nexus in MENA: spatial analysis"

_PeerJ, doi:10.7717/peerj.15708_

## Round 0.1 · original submission · Major Revisions

Please consider adding a schematic diagram to elaborate the study.

Reviewer 1 ·

Basic reporting

I enjoyed the reading of this paper. All sections of the paper are clearly written and the paper has a good structure to understand. However, the comments may help to raise the quality of the paper:
Methods in the abstract section should be discussed with more of the details.
A list of investigated countries should be added in the introduction section to convey idea of the sample of the study from the start of the paper.
The literature review section should be opened with the objective of the study and then the classification of the literature should be added. It will helpful to understand the other arguments about MENA countries in the introduction section.
The references style should be uni-form in the in-text references. For example, someplace 2-3 authors are written and et al. is used in the other places.

Experimental design

It is not clear that what are “i” and “t” in equation 1. Please define the sub-scripts.
Vi, ui, and wit in equation 2 should be describes to convey their meanings in the equation.

Validity of the findings

The discussing of the severity of oil rents and natural gas rents on the environment should be added in discussions of the results. I understand that oil rents and natural gas rents are part of income of the courtiers. However, to generate such income, some direct production activities of the oil rents and natural gas sectors could also have environmental problems.
The values of diagnostic tests should be written in the interpretation. So, the reader may understand that how a test accepts or rejects the hypothesis.

Additional comments

The limitations and future directions may add the value to the study. In this way, the chance of citation would also increase.

Reviewer 2 ·

Basic reporting

I thank the editor to give a chance to evaluate this paper. I have gone through the paper thoroughly and have the following concerns:
The policy implication discussed in the abstract section should be directly flowing from major results. The paper did not explore the diversification variable. Thus, the policy statement in the abstract section needs revisions and should be major findings of the analyses of the paper.
Lines 70-73 in the introduction section needs re-phrasing to make clear the context of arguments.
Line 78-79 in the introduction section about CO2 emissions as a global pollution emission needs a reference.
Lines 66-68 in the introduction section need more clarity to describe the definition of natural resource rents and their contribution in the income.
The word “they” in line 71 can be replaced with “the authors”.
In lines 229-230, the statement “However, the negative effect of NRR is also corroborated by a few studies.” needs references to be clear which studies found negative effects of NRR. The same references are needed for a positive effect of NRR.

Experimental design

The statement “most countries in it” in the methodology section in line 261 is not clear. Please make sure the region is in this context.
The function of weight (W) can be explained in more detail to understand the need for W in SDM equation 2.
The need to add Y and Y-square variables requires explanation and justifications. Please these variables to the EKC discussions.
The words “statistical findings” can be better to write statistical analyses before equation 2.
In lines 264-265, the SDM is suggested by Elhorst (2012). It requires an explanation of why SDM is suggested by Elhorst (2012) and why no other spatial models are suggested here.
Lines 271-273 should be re-phrased to make clear the hypotheses’ statements.

Validity of the findings

The format of cells in Table 1 should be alike and matched with the rest cells in Tables 1 & 2.
Tables of the results should be placed in the text. It could help to the readers for understand the interpretations by having a direct look at the results.
The conditions for the validity of the EKC could be added to the interpretations of the results. For instance, how it can be validated and how it cannot be confirmed.
In lines 324-325, please make clear how oil rents are significantly higher than natural gas rents. Please show some numerical values to support this argument.

Additional comments

The policy discussions may have space to increase. Moreover, the limitations and future directions can be added in the last lines of the conclusion section.

Reviewer 3 ·

Basic reporting

The title “Oil and natural gas rents and CO2 emissions nexus in MENA: spatial analysis” is an interesting research. However, there are some shortcomings in the paper, which must be correct before the next round of review. My specific comments are as follows:
Abstracts:
Authors needs to discuss the exact utilized spatial techniques in the method sub-section of the abstract section.
Introduction:
In the introduction section, “The present study is different altogether from the MENA region literature, which could only focus on the role of the consumption of energy on emissions.” is not clear that which study present or past could only focus on the role of the consumption of energy on emissions. Please make it more clear for readership.
Literature Review:
The abbreviation EU in the literature review must be written in full at its first use and then EU can be used afterwards.
The abbreviations like CO2, N2O, and CH4 should be written in full at their first use. Because, the readers might not understand the chemical titles of the gases.
The paragraph of the literature review section needs more explanation to discuss the literature gap and the contribution of the study.

Experimental design

Overall, method section is fine. But, following comments must be incorporated in the revised paper.
The method section should explain about the Environmental Kuznets Curve (EKC) and how it can be validated. The EKC is center of environmental literature now a days and it should have more detail discussion. Please also link how oil and natural gas rents would be helpful in the EKC model.
Please add the names of 17 MENA in the methodology section. These are given in discussions of figures but the readers need to know the full sample of the study in a single place of methods.
In the statement “Elhorst (2010) recommended the Wald test to verify the best spatial specification”, it is not clear what does mean by best spatial specification. It needs to be discussed in detail more in method section.
The statement “CO2 emissions are global emissions” in method section needs reference.

Validity of the findings

Results and discussions:
The heading of first column of Table 2 is missing. Please write an appropriate heading for this column.
The decimal points in Table 1 & 2 should be same for all results including P-value and coefficient of the results.
The interpretation of diagnostic tests in both tables 1 & 2 can be enhanced.

Additional comments

Conclusion:
In the conclusion section, the elasticity parameters should be discussed in more details in case of all major findings.

·

Basic reporting

Background of abstract section should have more powerful statement about the motivation of the study.
Method section in abstract section should clear sample countries under investigation. So, the reader can understand the scope of the paper from abstract section.
Keywords should add the region MENA.
The introduction section discussed the share of oil sector in GDP. In the same way, the share of natural gas should also be added to highlight the importance of natural gas sector in the investigated countries.
The statement in the introduction section “The maximum number of countries and time series from the MENA region are utilized as per the availability of data.” need to add sample countries in analysis.
Literature review section discuss only one study published in the year 2023. It should add further studies published in the year 2023 to highlight the importance of the topic in recent time.
The last concluding paragraph of literature review section should be added with references in justifying the literature gap and the contribution of the present study.

Experimental design

The first paragraph of method section mentions the nonlinear EKC. However, the EKC needs more discussions as reader is unaware of the EKC and authors should make it clear that what is the EKC and how it can be tested and what is importance of its testing in the sample MENA countries.
The equation 2 explains the utilized SDM specification. However, the SDM is compared with SAR and SEM. Therefore, the equations of SAR and SEM should also be discussed in the methods to show a complete picture of methodology.
The justification of using spatial analyses should be expanded in the context of the sample countries.
Sub-scripts of the variables and error terms of the equations are not discussed in the methodology. Thus, the equations need more explanations.

Validity of the findings

In the statement “Table 1 exposes the results of FE without spatial dimensions. At first, we apply the LR test on FE and the results reject the H0 for FE-country and FE-both specifications. So, both are valid and superior over pooled regression. However, the LR test could not reject H0 for FE-time.”, how LR test accept or reject the H0 need more clarity of results behind it in the discussions.
The statement “Thus, FE models are biased as spatial autocorrelation is substantiated in all models.” need reference.
In the statement “Then, we apply LM and LR tests on both models to test the suitability of the SDM.”, table 2 did not contain the LM test. However, LR and the Wald tests are presented. So, the statement need correction or more explanations.
The effects of oil rents are significant and natural gas rents are not. These results need the background facts responsible for such results.
The discussions on the results of weighted variables are ignored in interpretation.
Please provide more explanation of tables and summarize the insights.

---

## Round 0.2 · accepted · Accept

I have gone through the manuscript. authors have substantially revised the manuscript. Therefore, I recommend that manuscript can be accepted.

·

Basic reporting

The suggestions have been incorporated

Experimental design

The suggestions have been incorporated

Validity of the findings

The suggestions have been incorporated

Additional comments

The suggestions have been incorporated.
I recommend the acceptance